# Inhibition of Key Enzymes Linked to Obesity and Cytotoxic Activities of Whole Plant Extracts of *Vernonia mesplilfolia* Less

**Jeremiah Oshiomame Unuofin** [1,2,*,†] **, Gloria Aderonke Otunola** [2] **and Anthony Jide Afolayan** [2]

1   Department of Life and Consumer Sciences, University of South Africa, Cnr Christiaan de Wet and Pioneer Ave, Private Bag X6, Florida 1710, South Africa

2   Department of Botany, Medicinal Plants and Economic Development (MPED) Research Centre, University of Fort Hare, Alice 5700, South Africa; gotunola@ufh.ac.za (G.A.O.); aafolayan@ufh.ac.za (A.J.A.)

*   Correspondence: Unuofjo@unisa.ac.za; Tel.: +27-73-806-7879

†   J.O. Unuofin, Department of Life and Consumer Sciences, University of South Africa, Cnr Christiaan de Wet and Pioneer Ave, Private Bag X6, Florida, 1710, South Africa.

**Abstract:** The whole plant of *Vernonia mespilifolia* is widely used as a traditional remedy for obesity in South Africa. The aim of this study was to investigate the anti-obesity and cytotoxic effects of *Vernonia mespilifolia* extracts *in vitro*. The α-amylase, α-glucosidase, and lipase inhibitory activities of aqueous and ethanol extracts of *Vernonia mespilifolia* were investigated, while the cytotoxic effects of these extracts were analyzed using Hoechst 33342 and propidium iodide (PI) dual staining on a human cervical HeLa cell line. The results showed that the $LC_{50}$ (the concentration of a material will kill 50% of test organisms) values of aqueous and ethanol extracts of *Vernonia mespilifolia* were >200 and 149 µg/mL, respectively, to HeLa cells. Additionally, the ethanol extract exhibited the strongest inhibitory effect on the pancreatic lipase (Half-maximal inhibitory concentration ($IC_{50}$) = 331.16 µg/mL) and on α-amylase ($IC_{50}$ = 781.72 µg/mL), while the aqueous extract has the strongest α-glucosidase ($IC_{50}$ = 450.88 µg/mL). Our results suggest that *Vernonia mespilifolia*'s acclaimed anti-obesity effects could be ascribed to its ability to inhibit both carbohydrate and fat digesting enzymes.

**Keywords:** *Vernonia mespilifolia*; α-amylase; α-glucosidase; lipase; HeLa; Hoechst 33342/PI staining

## 1. Introduction

Obesity is a multifactorial metabolic disorder that is brought about by western dietary patterns, genetic makeup, lifestyle patterns and a host of environmental factors [1]. The global prevalence of this metabolic disorder has witnessed a twofold increment since 1980 to a level whereby one-third of the world population is regarded as being overweight or obese [2,3]. Obesity has been tagged as a serious public health challenge that is interlinked with a number of other metabolic disorder such as heart-related diseases, diabetes, cancer, arthritis and a host of other complications [4]. The socio-economic burden of this disorder on an individual is one of the contributory factors of the onset of depression [5]. The financial stress that the disorder brings on the individual and the government is enormous. According to the United State government in 2003, 75 billion dollars were expended on the management and treatment of obesity and its comorbidity, and it has been forecasted that this amount will rise by 130% come 2030 [6,7].

Over the years, a myriad of conventional drugs (lorcaserin, naltrexone–bupropion, and sibutramine) have been administered for the management and treatment of obesity [8], but their

numerous side effects (anxiousness, fatigue, flatulence, headache and insomnia) have caused them to be withdrawn from the market [6,9,10]. Therefore, plant-derived medicines in the form of herbal remedies and natural products have gained attention as first lines of protection in public health for contending with diseases and their complications [11]. Though these herbal remedies are enriched with diverse bioactive constituents with therapeutic efficacy for the management of obesity and other comorbidities, the presence of toxic materials in these remedies could be a demerit to their usage [10,12]. It is thus essential to find out the toxicity levels of herbal remedies in order to ascertain their safety. It is of grave importance to determine the cytotoxic potential of medicinal plants in order to determine their safe limits.

*Vernonia mespilifolia*, a shrub that belongs to the Asteraceae family, is naturally distributed all over South Africa [13,14]. Its ethnomedicinal potential spans from the curing of heartwater diseases (in ruminant animals like goat), high blood pressure, and the regulation of body mass [15–17]. Bioactive components in oil extracted from this plant possess a number of pharmacological activities against cancer, inflammation, microbes and high blood pressure [18]. Furthermore, a number of solvent (organic and inorganic) extracts of *V. mespilifolia* are rich sources of anti-oxidant, -microbial, -mycobacterial and -trypanosomal activities [19–22]. The study is therefore aimed at investigating some *in vitro* anti-obesity activities of *V. mespilifolia* based on ethnomedicinal claims for its use in the treatment of obesity [17] and evidence of reported antidiabetic and cytotoxic activities reported for its closely related species [23–25]. We employed different *in vitro* assays involving the inhibition of carbohydrate and lipid metabolizing enzymes, as well as the HeLa cell line in order to establish preliminary data on the potential cytotoxic activities of the aqueous and ethanol extracts of *V. mespilifolia*.

## 2. Results

### 2.1. Inhibition of Amylase, Glucosidase and Lipase by V. mespilifolia

The inhibitory effects of aqueous and ethanolic extracts of *V. mespilifolia* on three digestive enzymes linked with type 2 diabetes and obesity are depicted in Figures 1–3 and Table 1. The pancreatic amylase inhibitory potential possessed by both extracts was mild, as was observed in a concentration-dependent pattern. The ethanolic extract had a better inhibitory capacity than the aqueous extract, as is shown with its lesser Half-maximal inhibitory concentration ($IC_{50}$) value that is presented in Table 1. At the highest investigated concentration (200 µg/mL), both extracts (aqueous and ethanolic) demonstrated a substantial inhibition of amylase by 21.76% and 24.21%, respectively. Nonetheless, the acarbose inhibition of the amylase enzyme (92.97%) surpassed that of both extracts (Figure 1).

The inhibition of glucosidase by both extracts (aqueous and ethanolic) is presented in Figure 2. Both extracts' inhibitory potential on this enzyme followed a similar pattern to that of their amylase activity. The aqueous extract presented a higher inhibition of α-glucosidase than the ethanolic extract; this is shown in Table 1. At the highest investigated concentration (200 µg/mL), both extracts (aqueous and ethanolic) demonstrated a substantial inhibition of glucosidase by 28.99% and 29.88%, respectively. Nonetheless, the EGCG (epigallocatechin gallate) inhibition of the glucosidase enzyme (67.57%) surpassed that of both extracts (Figure 2).

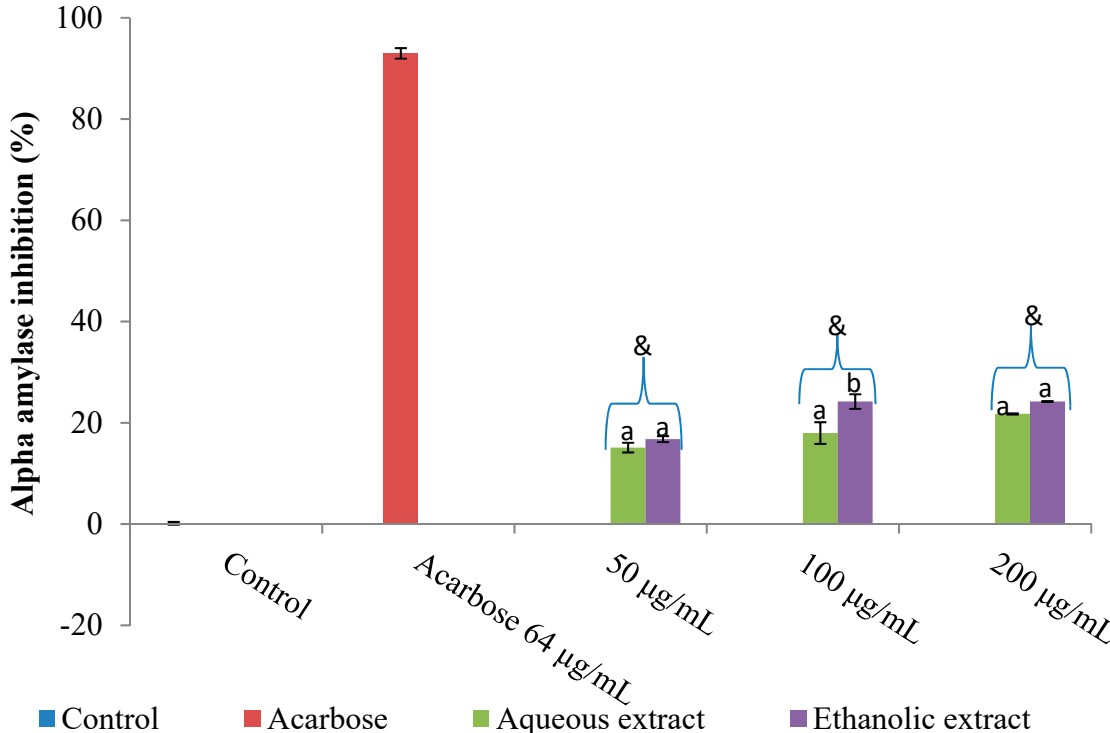

**Figure 1.** Bar chart showing the inhibition of α-amylase by aqueous and ethanolic extracts of *Vernonia mespilifolia*. The values in the bar chart are expressed as mean (α-amylase inhibition) ± standard deviation (SD) (n = 5). Mean separation was done by Least Significant Difference (LSD) ($p < 0.05$). Sets of bars (the same concentration) with different alphabets are significantly different; $p < 0.05$. [&] Less than acarbose (positive control).

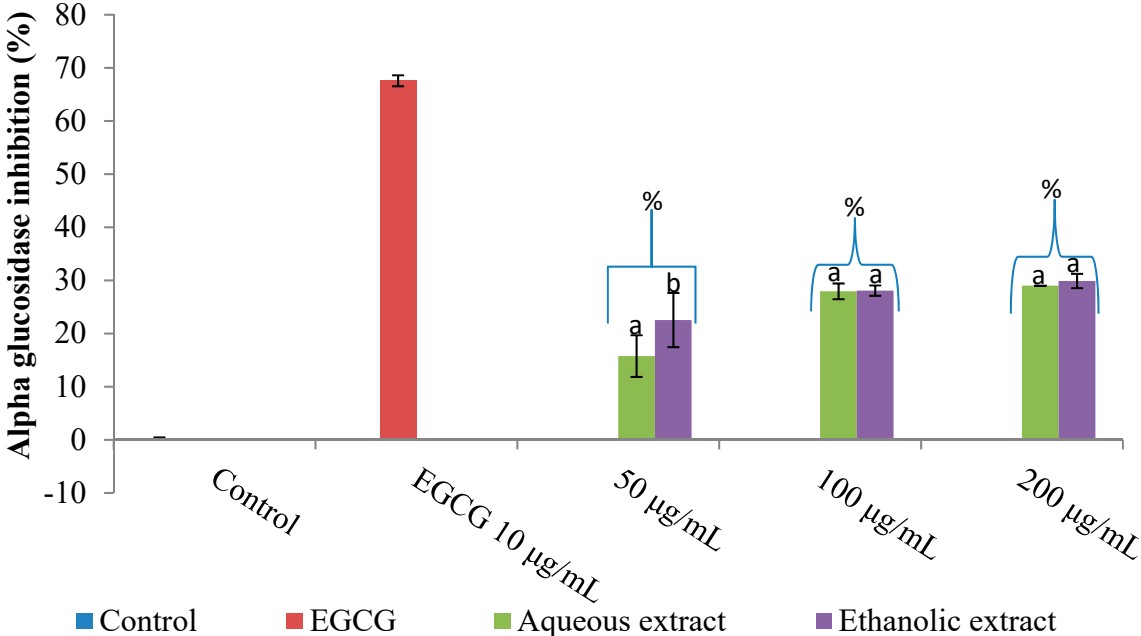

**Figure 2.** Bar chart showing the inhibition of α-glucosidase by aqueous and ethanolic extracts of *Vernonia mespilifolia*. The values in the bar chart are expressed as mean (α-glucosidase inhibition) ± SD (n = 5). Mean separation was done by LSD ($p < 0.05$). Sets of bars (the same concentration) with different alphabets are significantly different; $p < 0.05$. [%] Less than the EGCG (epigallocatechin gallate) (positive control).

In the case of lipase inhibition by both extracts of *V. mespilifolia*, we used tetrahydrolipstatin (orlistat) as a positive control (Figure 3). Both extracts demonstrated a nonsignificant inhibition of lipase activity at their concentration range. At the highest investigated concentration (200 µg/mL), the extracts' (aqueous and ethanolic) inhibitory activity was 13.57% and 17.88%, respectively, which was far less in relation to tetrahydrolipstatin (orlistat) (72.98%).

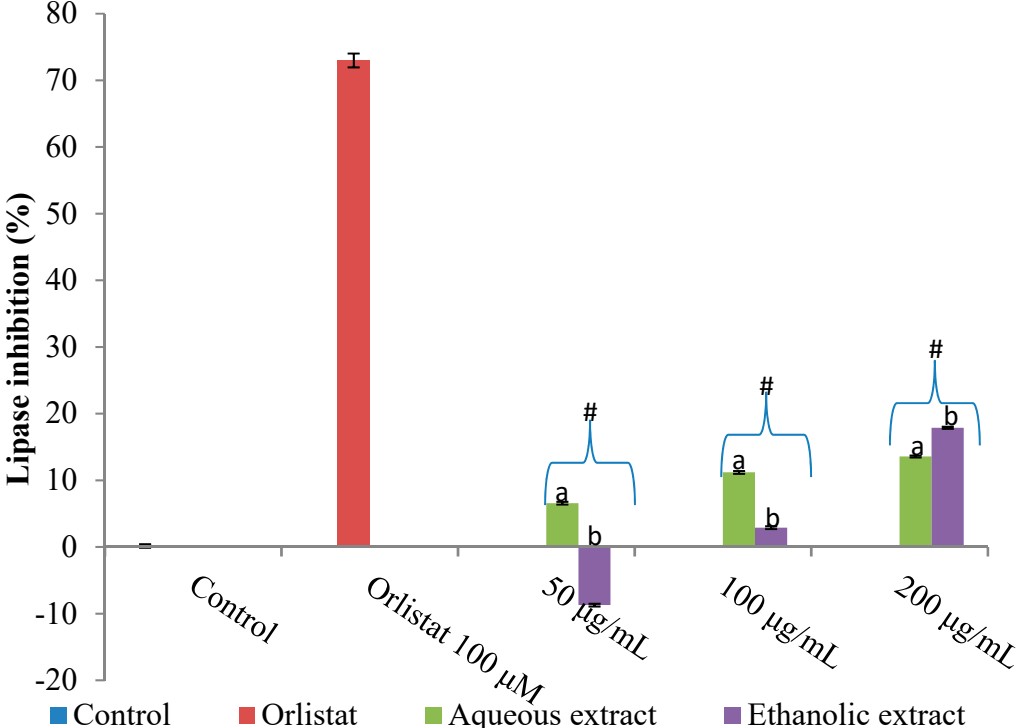

**Figure 3.** Bar chart showing the inhibition of the pancreatic lipase by aqueous and ethanolic extracts of *Vernonia mespilifolia*. The values in the bar chart are expressed as mean (lipase inhibition) ± SD (n = 5). Mean separation was done by LSD ($p < 0.05$). Sets of bars (the same concentration) with different alphabets are significantly different; $p < 0.05$. [#] Less than the Orlistat (positive control).

**Table 1.** Inhibitory potential (Half-maximal inhibitory concentration ($IC_{50}$) values) of aqueous and ethanolic extracts of *Vernonia mespilifolia*.

| Name of Extract | $IC_{50}$ Values (µg/mL) | | |
| --- | --- | --- | --- |
| | *α*-Amylase | *α*-Glucosidase | Lipase |
| Aqueous | 847.49 ± 1.69 | 450.88 ± 2.84 | 1028.18 ± 3.27 |
| Ethanolic | 781.72 ± 2.58 | 637.35 ± 1.89 | 331.16 ± 1.85 |

Data are presented as mean ± SD values of five replicate determinations.

Both extracts (aqueous and ethanolic) of the plant displayed inhibitory activity against *α*-amylase ($IC_{50}$ 847.49 ± 1.69 and 781 ± 2.58 µg/mL, respectively) and *α*-glucosidase ($IC_{50}$ 450.88 ± 2.84 and 637 ± 1.89 µg/mL, respectively). In addition, the $IC_{50}$ values of both extracts (aqueous and ethanolic) of *V. mespilifolia* $IC_{50}$ for lipase inhibition were 1028.18 ± 3.27 and 331.16 ± 1.85 µg/mL, respectively (Table 1).

### 2.2. Cytotoxicity Activity

Our finding on the cytotoxic potential of both extracts against the HeLa cervical cell line was carried out through the dual staining technique (Hoechst 33342/PI). Figure 4 shows the fluorescent pictographs of both the treated and untreated cells. The cytotoxic potential of the ethanolic extract

was illustrated with a large number of death cells, which was characterized by a critical decline in cell numbers (blue color) with an upsurge amount of lifeless cells (bright red color) in comparison to the other two samples (aqueous extract and control/untreated).

Figure 5a,b reveals both extracts' (aqueous and ethanolic) cytotoxic potentials against cervical cell lines (HeLa), as presented by the staining techniques used in this study. Our findings revealed that, unlike the ethanolic extract, there was no considerable amount of cell death in across all concentrations for the aqueous extract. At 100 and 200 µg/mL of the ethanolic extract, 27.15% and 73.24% cell death, respectively, were recorded. The $IC_{50}$ value for ethanolic extract was 149.12 µg/mL, whereas that of the aqueous extract was far higher than 200 µg/mL (Table 2).

**Table 2.** Cytotoxic activity ($LC_{50}$ values) of the aqueous and ethanolic extracts of *Vernonia mespilifolia*.

| Name of Extract | Cytotoxicity $IC_{50}$ (µg/mL) |
|---|---|
| Aqueous | >200 |
| Ethanolic | 149.12 ± 3.82 |

Data are presented as mean ± SD values of five replicate determinations.

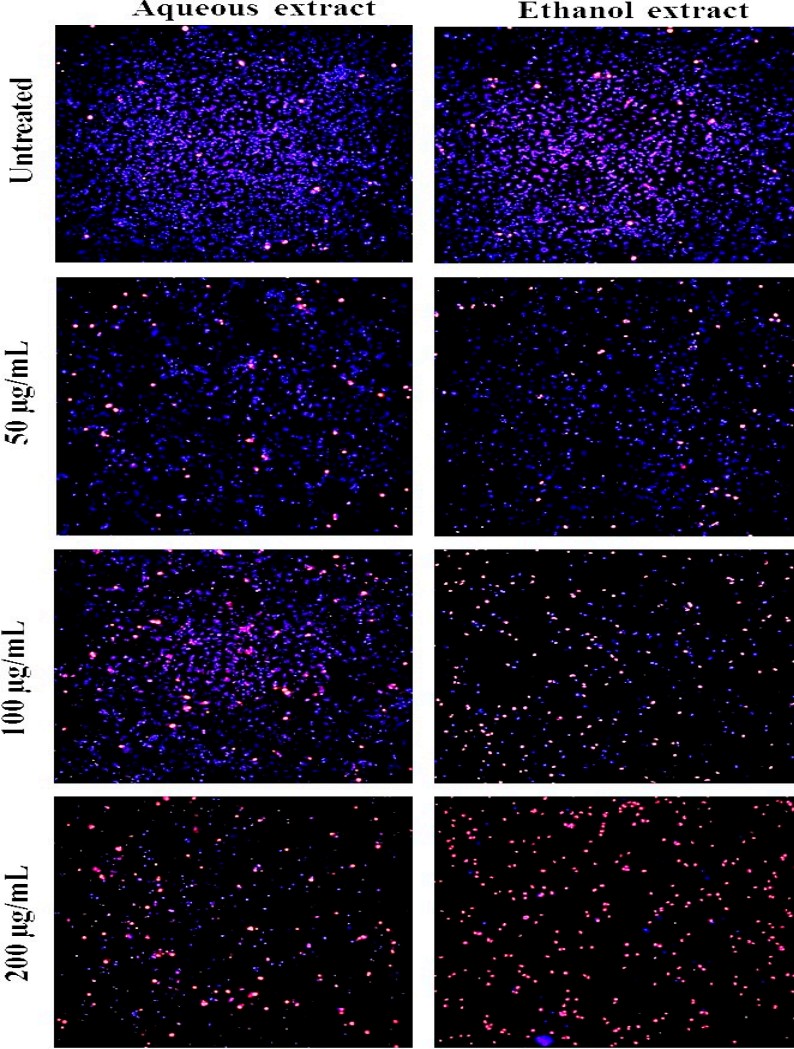

**Figure 4.** Representative images of aqueous and ethanolic extracts of *V. mespilifolia* at the varying concentrations on the HeLa cells after 48 h exposure using Hoechst 33,342 (blue) and propidium iodide (red) dual staining.

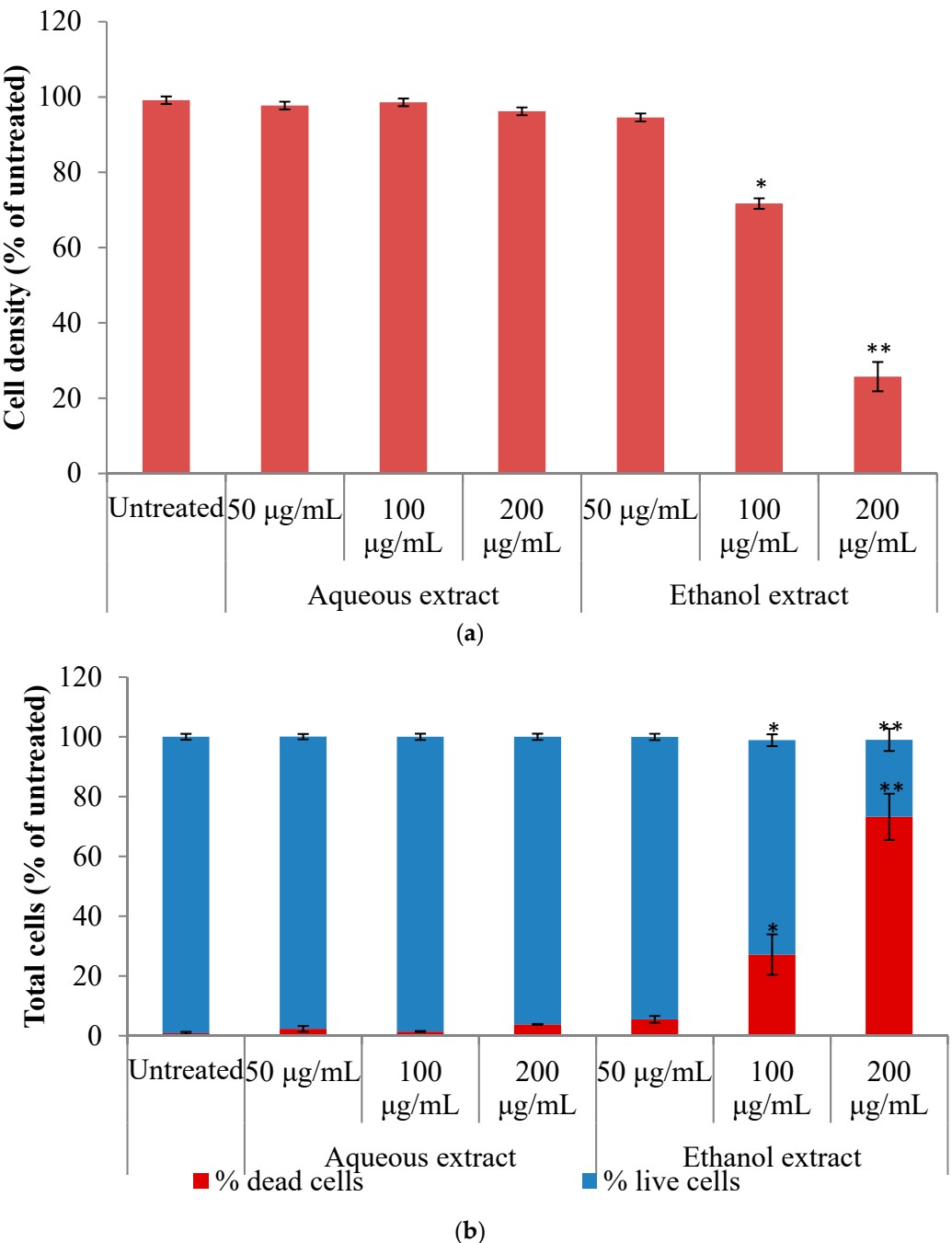

**Figure 5.** (**a**) The number of live HeLa cells present after 48 h of exposure to aqueous and ethanolic extracts of *Vernonia mespilifolia*. Nuclei were stained with Hoechst 33,342 and PI (propidium iodide), and images were acquired on the ImageXpress Extended Light Scatterer (XLS) Micro widefield microscope. The total number of nuclei (cells) were counted using the Multiwavelength Cell Scoring module. Data points represent mean ± SD of five quintuplicate values (* $p < 0.05$; ** $p < 0.01$ compared to untreated sample). (**b**) The total number of HeLa cells (live + dead) present after 48 h of exposure to aqueous and ethanolic extracts of *Vernonia mespilifolia*. Nuclei were stained with Hoechst 33,342, and images were acquired on the ImageXpress XLS Micro widefield microscope. The total number of nuclei (stained with Hoechst) and the number of dead nuclei (stained with Hoechst and PI) were counted using the Multiwavelength Cell Scoring module. Live cells were stained with Hoechst only, while dead cells were stained with both dyes. Data points represent mean ± SD of five quintuplicate values (* $p < 0.05$; ** $p < 0.01$ compared to untreated sample).

## 2.3. Discussion

The reliance of mankind on herbal remedies in the form of concoction, decoction, or other forms as substitute/alternative therapies for synthetic drugs is growing on an exponential basis. These remedies are mostly a combination of more than one natural plant product [26]. The increased abuse or indiscriminate consumption of these natural products without proper technical authentication for their capability and healthiness for consumers is of paramount importance. In light of these, our study aimed to evaluate the in vitro inhibitory potential of enzymes associated with obesity, as well as the cytotoxicity potential of the aqueous and ethanolic extracts of *V. mespilifolia*. According to [27], the discovery and development of inhibitors (nutrient absorption and digestion) are promising targets for the management and treatment of metabolic disorders such as obesity and diabetes. For example, acarbose is a pseudotetrasaccharide approved by the Food and Drug Administration (FDA) for the treatment of adults with type 2 diabetes mellitus as an adjunct to diet only or diet and exercise, depending on the patient's health status. Acarbose is a complex oligosaccharide that acts as a competitive, reversible inhibitor of pancreatic alpha-amylase and membrane-bound intestinal alpha-glucoside hydrolase. It delays the digestion of carbohydrates and slows down glucose absorption, resulting in a reduction of postprandial glucose [28,29]. Epigallocatechin gallate (EGCG) and other polyphenols isolated from green tea lower the level of blood glucose and thus have anti-diabetic effects. Until recently, little information was known about the interaction between acarbose, an effective first-line drug which has inhibitory activities against α-amylase and α-glucosidase, and green tea [30,31].

The restriction in the activity of α-amylase and α-glucosidase are beneficial targets within type 2 diabetes therapy owing to causing a delay in making available blood sugar [32]. Herbal remedies that possess an effective inhibitory capacity against glucosidase are more vital in restricting the circulation of glucose during carbohydrate metabolism in comparison to the inhibitory strength of amylase. Consequently, a herbal remedy that exerts both fair α-amylase and robust α-glucosidase inhibition capacities could serve as a promising curative stratagem in the area of reducing easy carbohydrate metabolism, which brings about influx glucose in the body of diabetic subjects [10,33]. The ultimate demerits of synthetic medication with robust amylase and glucosidase inhibitory capacities is that they possess the ability to bring about the unwarranted hindrance of amylase activity, which could promote anomalous digestive problems (the fermentation of unbroken down sugar products) that are associated with the colon [10,34].

Our findings highlight that there was an increase in the inhibition of both amylase and glucosidase enzymes in a concentration-dependent manner (Figures 1 and 2). Generally, both extracts had a moderate inhibition against amylase; however, against α-glucosidase, the aqueous extract displayed a superior inhibitory capability when compared with the ethanol extract (Table 1). The breakdown of dietary lipids and their assimilation in the intestine is predominantly accomplished via the pancreatic lipase [35].

The inhibition of this enzyme serves as a controlling point in overweight/obesity therapy. According to [36], there is a strong correlation between insulin sensitivity and an upsurge in body lipid composition, thus promoting the importance of pancreatic lipase inhibition as an advantageous target in adjusting lipid absorption in the body. Plant materials with abundant biological active compounds such as phenolic acid, stilbenes, anthocyanidins and lignans have been linked to a lowering of pancreatic lipase activity [37,38]. These biologically active compounds elicit their inhibitory activity by a non-specific binding to the aforementioned enzyme [39]. In our previous study on the phytochemical content of *V. mespilifolia*, it was observed that the ethanolic extract had a larger phytochemical content (flavonoids, proanthocyanidins and tannins) in comparison to the aqueous extract [22]. Furthermore, the varying inhibitory potential of both extracts on the digestive enzymes shows their therapeutic concentrations. Therefore, it is important to recognize the toxic levels of these identified therapeutic doses. Our findings revealed that at a certain concentration, the ethanolic extract showed some level of toxicity on the HeLa cells. Some researchers [10,40] have reported that the US National Cancer Institute (NCI) Plant Screening scheme has shown that any crude extract incubated for a period of 48–72 h with

$LC_{50} \geq 20$ μg/mL ($LC_{50}$ is the lethal concentration required to kill 50% of test organisms) is safe or cytotoxic if its $LC_{50}$ value is less than 20 μg/mL. From our results, it can be inferred that both extracts had $LC_{50} > 20$ μg/mL and may be regarded as orally safe for the management of obesity. According to [41], plants containing toxic compounds could possess potent biological activities, inasmuch as toxicity at low doses can be linked to their pharmacological activity. In addition, it is noteworthy that cytotoxicity noticed *in vitro* is not always greeted in vivo. This could be owed to the fact that within a biological system, a number of toxic compounds are capable of undergoing biotransformation, thus bringing about the formation of end products with minute toxic levels [42,43].The toxic character of plant extracts are mostly a pointer to factors like the season of harvest of the plant material, the presence of heavy metals in the soil, the growth stage of the plant, and a host of other factors [44].

## 3. Experimental Procedure

### 3.1. Chemicals, Reagents and Equipment

95% ethanol, yeast α-glucosidase, p-nitrophenyl-α-D-glucopyranoside (pNPG), tetrahydrolipstatin (orlistat), porcine pancreatic lipase, porcine pancreatic amylase, epigallocatechin gallate (EGCG), acarbose, and other chemicals/reagents were all acquired from Merck and Sigma-Aldrich in Germany. Büchi Rotavapor (Büchi, Switzerland) and a multi-mode reader (DAR 800, USA) were used.

### 3.2. Plant Materials

The whole plant part of *V. mespilifolia* Less. was obtained from Zihlahleni Village located in Eastern Cape, South Africa, mid-2015. The sample was then washed and oven-dried at 40 °C for 3 days to a constant mass; following this, the samples were powdered by means of a blender, and they were then stored in zip-lock bags.

### 3.3. Extraction Procedure

One-hundred-and-fifty grams of ground sample were extracted separately using 1.5 L of ethanol and water respectively, and they were then placed on a shaker for 48 h. After filtering with the aid of Whatman No. 1 filter paper, the ethanolic extract was concentrated in vacuo rotary evaporator. The aqueous extract was lyophilized with a freeze dryer.

### 3.4. Inhibitory Potential of Plant Extracts

The inhibition of the alpha-amylase enzyme in vitro by both solvent extracts was carried out in accordance with the published method of [45] with minor modifications. Briefly, acarbose (64 μg/mL) and 30 μL of the extracts (1 mg/mL stock solution) at changeable concentrations (50–200 μg/mL) were incubated for 10 min at 37 °C with 10 μL of the enzyme. Thereafter, 40 μL of 1% starch was added and incubated more for 30 min at 37 °C. Twenty microliters of 1 molar hydrochloric acid and 75 μL $I_2$ reagent were introduced into the resultant solution to put a to end to the reaction, and absorbance was taken at 580 nm. Acarbose, a known α-amylase inhibitor, was used as a positive control. Additionally, controls without both the enzyme inhibitor (phosphate buffer) and substrate (starch) were also included in the assay in order to ascertain that no reaction occurred when either the enzyme or substrate was absent. This brought about the elimination of false-positive results.

Inhibition activity (%) = [1 − (Abs of test sample $_{580\,nm}$/Abs of enzyme control $_{580\,nm}$)] × 100

### 3.5. Alpha-Glucosidase

The inhibition of alpha-glucosidase enzyme in vitro by both solvent extracts was carried out in accordance with the published method of [46] with minor modifications. Briefly, 40 μL of 50 μg/mL alpha-glucosidase enzyme were preincubated for 5 min at 37 °C with the addition of 10 μL of extract/EGCG. Thereafter, 10 μL of a 5 mmol $L^{-1}$ p-nitrophenyl-α-D-glucopyranoside (pNPG) solution

was then introduced into the reaction mixture and was further incubated for 30 min at 37 °C. The resulting reaction was terminated with the addition of 50 μL of $Na_2CO_3$, and absorbance was read at 405 nm. Epigallocatechin gallate (EGCG), a known α-glucosidase inhibitor, was used as a positive control. The controls without the enzyme inhibitor (phosphate buffer) and without the substrate (pNPG) were also included in the assay.

Inhibition activity (%) = [1 − (Abs of test sample $_{405\ nm}$/Abs of enzyme control $_{405\ nm}$)] × 100

*3.6. Pancreatic Lipase*

The inhibition of the pancreatic lipase enzyme in vitro by the aqueous and ethanolic extracts of *V. mespilifolia* was carried out in accordance with the published method of [47] with minor modifications using 96-well microplate. Firstly, 10 μL of the extracts (1 mg/mL stock solution) at changeable concentrations (50–200 μg/mL) and tetrahydrolipstatin (100 μM) were incubated for 15 min at 37 °C with 40 μL of the enzyme (pancreatic lipase (50 mg/mL). Thereafter, 170 μL of the substrate (10 mM pNPB) was added, and this was followed by an incubation period for a quarter of an hour at 37 °C. The resulting mixture absorbance was taken at 405 nm. Orlistat, a known lipase inhibitor, was used as a positive control. Controls without the enzyme inhibitor (phosphate buffer) and without the substrate (pNPB) were also included in the assay.

Inhibition activity (%) = [1 − (Abs of test sample $_{405\ nm}$/Abs of enzyme control $_{405\ nm}$)] × 100

## 4. Cell Cytotoxicity Assay Using the Hoechst/PI Staining

*4.1. Cell Line*

HeLa cells were procured from American Type Culture Collection (ATCC) (Rockville, MD, USA). Cell growth was achieved at 37 °C in a humidified incubator with 5% CO2 and 95% air atmosphere. Dulbecco's modified eagle medium (DMEM) was complemented with 10% (v/v) inactivated bovine serum, penicillin G, and streptomycin (100 mg/L). The cytotoxicity activity was measured according to a published method [10] with slight modifications.

% cell density = A570 nm of treated cells/A570 nm of untreated cells × 100

*4.2. Data Evaluation*

All assays were conducted in five replicates, and the obtained results (mean ± SD) were statistically analyzed using MINITAB 17 and a one-way ANOVA by Fischer's least significant difference (LSD) to determine significant differences in all the parameters. Values were considered statistically significant at $p < 0.05$.

## 5. Conclusions

The presence of phytochemical compounds that can exercise the inhibition of digestive enzyme-linked carbohydrate and lipid metabolism in *V. mespilifolia* is revealed in our study. These results suggest that both aqueous and ethanol extract possess inhibitory activity against a number of crucial enzymes associated with diabetes and obesity therapy.

**Author Contributions:** Investigation, J.O.U., G.A.O., and A.J.A.; supervision, G.A.O., and A.J.A.; writing—original draft, J.O.U.; writing—review and editing, J.O.U., and G.A.O.

**Funding:** This research was funded by the Govan Mbeki Research Development Centre (GMRDC), University of Fort Hare, South Africa (grant number C127) and The APC was funded by University of Fort Hare.

**Acknowledgments:** Govan Mbeki Research Development Centre (GMRDC), University of Fort Hare, South Africa.

**Conflicts of Interest:** No conflict of interest was disclosed by the authors.

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
