# Peer review of "Inhibition of Key Enzymes Linked to Obesity and Cytotoxic Activities of Whole Plant Extracts of Vernonia mesplilfolia Less"

_processes, doi:10.3390/pr7110841_

Round 1

Reviewer 1 Report

I understand that the authors want to say the ethanol extract of Vernonia mesplilfolia is not toxic based on the criteria by the American National Cancer Institute. However, still remain the problem to mention that the ethanol extract of Vernonia mesplilfolia is useful for anti-obesity. Because, the IC50 value of the ethanol extract for cytotoxicity were 150 µg/mL that is lower than other IC50 values for enzymatic activities in Table1. These results suggest that the ethanol extract is not promising to use for anti-obesity. How do the authors explain?

The explanation of evidence of reported antidiabetic and cytotoxic activities reported for its closely related species to Vernonia mesplilfolia should be described in Introduction.

Author Response

Point 1: I understand that the authors want to say the ethanol extract of Vernonia mespilifolia is not toxic based on the criteria by the American National Cancer Institute. However, still remain the problem to mention that the ethanol extract of Vernonia mespilifolia is useful for anti-obesity. Because the IC50 value of the ethanol extract for cytotoxicity was 150 µg/mL that is lower than other IC50 values for enzymatic activities in Table 1.

Response:

We totally get your view about the supposed toxic nature of the ethanol extract of Vernonia mespilifolia however we like to stand by the criteria set by the American National Cancer Institute and also show you some publications/dissertations with similar findings/results as ours. In fact, in one of the below references (Nolitha et al. 2011), the authors went ahead to isolate compounds from an extract whose IC50 for cytotoxicity was way lower than its anti-diabetes activity. Below are some of the references with similar findings.

Gupta et al. 2014. In-vitro cancer cell cytotoxicity and alpha-amylase inhibition effects of seven tropical fruit residues. Asian Pacific Journal of Tropical Biomedicine, 4, S665-S671. Nolitha NK et al. (2011). Antidiabetic activity of Terminalia sericea Burch. Ex DC constituents. (Master dissertation). University of Pretoria, South Africa. Pages 88-92. Asong JA et al. 2019. Antimicrobial activity, antioxidant potential, cytotoxicity and phytochemical profiling of four plants used against skin diseases. Plants, 8(9), 350 

Reviewer 2 Report

In the revised manuscript entitled “Inhibition of key enzymes linked to obesity and cytotoxic activities of whole plant extracts of Vernonia mesplilfolia Less- A promising medicinal crop in South Africa” authors have addressed some of the points raised by reviewers. But, the manuscript should be modified thoroughly in order to get published in Processes.

The following points should be addressed before accepting the manuscript.

The manuscript should be thoroughly modified to avoid grammatical errors The manuscript should eb modified to describe each and every experiments that the authors performed with the observation followed by its meaning/interpretation. For example, in the results, it is written that “the inhibitory effects of aqueous and ethanolic extracts of V. mespilifolia on three digestive enzymes linked with type 2 diabetes and obesity are depicted in Fig. 1-3 and Table 1”, but the figure 1 is not explained separately in the manuscript. Authors should explain each and every figures thoroughly in the text. I have already asked about the “Control” in each graph, what is the control used? these should be added to the manuscript text Authors gave detailed explanation for reviewers questions. I recommend the authors to include these explanations in the manuscript and add proper references to make the paper more understandable to readers.

Overall, the manuscript in the present form should be modified thoroughly to explain each and every data in detail.

Author Response

Point 1: The manuscript should be thoroughly modified to avoid grammatical errors.

Response:

The entire manuscript has been thoroughly read and all grammatical errors corrected

Point 2: The manuscript should be modified to describe each and every experiment that the authors performed with the observation followed by its meaning/interpretation.

Response:

Each experimental results in the manuscript have been explained clearly.

Point 3: I have already asked about the "control" in each graph, what is the control used? These should be added to the manuscript text. The authors gave a detailed explanation for the reviewers' questions. I recommend the authors to include these explanations in the manuscript and add proper references to make the paper more understandable to readers.

Response: 

The authors have followed your recommendation of inserting the controls used in each experiment in-text for the purpose of clarity and understanding of the readers

Round 2

Reviewer 1 Report

Comments,

Descriptions in abstract, the authors can rewrite below.

1). Page 1, lines 18-19, "The results showed that both extracts were found non-toxic to HeLa cells"

possible change to

"LC50 values of aqueous and ethanol extracts of Vernonia mespilifolia were >200 and 149 µg/ml, respectively."

2). Page 1, lines 21-23, "Our results suggest that Vernonia mespilifolia is non-toxic and its acclaimed anti-obesity effects could be ascribed to its ability to inhibit both carbohydrate and fat digesting enzyme in the body."

possible change to

"Our results suggest that Vernonia mespilifolia acclaimed anti-obesity effects could be ascribed to its ability to inhibit both carbohydrate and fat digesting enzyme."

I understand that the authors criteria, which set by the American National Cancer Institute. That criteria is applicable to cancer research and useful for 1st screening of anti-cancer agent candidates. In your research, ED50 values of the ethanol extract of Vernonia mespilifolia were higher than LD50 value. Therefore, the fraction is not useful for health because of the cytotoxicity. After purification of an active substance, it is possible to use for health, if the toxic compounds in the ethanol extract of Vernonia mespilifolia will be removed.

Author Response

Response to Reviewer 1 Comments

Descriptions in the abstract, the authors can rewrite below.

1). Page 1, lines 18-19, "The results showed that both extracts were found non-toxic to HeLa cells"

possible change to

"LC50 values of aqueous and ethanol extracts of Vernonia mespilifolia were >200 and 149 µg/ml, respectively."

Response:  The authors have incorporated your suggestion into the abstracts section of the manuscript

2). Page 1, lines 21-23, "Our results suggest that Vernonia mespilifolia is non-toxic and its acclaimed anti-obesity effects could be ascribed to its ability to inhibit both carbohydrate and fat digesting enzyme in the body."

possible change to

"Our results suggest that Vernonia mespilifolia acclaimed anti-obesity effects could be ascribed to its ability to inhibit both carbohydrate and fat digesting enzyme."

Response: The authors have incorporated your suggestion into the abstracts section of the manuscript

I understand that the authors criteria, which set by the American National Cancer Institute. That criteria is applicable to cancer research and useful for 1st screening of anti-cancer agent candidates. In your research, ED50 values of the ethanol extract of Vernonia mespilifolia were higher than LD50 value. Therefore, the fraction is not useful for health because of the cytotoxicity. After purification of an active substance, it is possible to use for health, if the toxic compounds in the ethanol extract of Vernonia mespilifolia will be removed.

Response: According to [41], plants containing toxic compounds could possess potent biological activities, inasmuch as toxicity at low doses can be linked to their pharmacological activity. In addition, it is noteworthy that cytotoxicity noticed in vitro is not always greeted in vivo. This could be owing to the fact that within the biological system, a number of toxic compounds are capable of undergoing biotransformation thus bringing about the formation of end products with minute toxic levels [42, 43].

McGaw, L.J.; van der Merwe, D.; Eloff, J.N. In vitro anthelmintic, antibacterial and cytotoxic effects of extracts from plants used in South African ethnoveterinary medicine. Vet J2007

Nchu, F.; Githiori, J.B.; McGaw, L.J.; Eloff, J.N. Anthelmintic and cytotoxic activities of extracts of Markhamia obtusifolia Sprague (Bignoniaceae) Vet Parasitol2011

Kudumela, R.G.; McGaw, L.J.; Masoko, P. Antibacterial interactions, anti-inflammatory and cytotoxic effects of four medicinal plant species. BMC Complementary and Alternative Medicine 2018.

Reviewer 2 Report

Authors have addressed all the comments raised by the reviewer. The manuscript can be accepted in the present form. 

Author Response

The reviewer has accepted the manuscript in this format.

Round 3

Reviewer 1 Report

The authors responded to my comments.

This manuscript is a resubmission of an earlier submission. The following is a list of the peer review reports and author responses from that submission.

Round 1

Reviewer 1 Report

The manuscript entitled “Inhibition of key enzymes linked to obesity and cytotoxic activities of whole plant extracts of Vernonia mesplilfolia Less- A promising medicinal crop in South Africa” authored by J. O. Unuofin and co-workers describes the evaluation of the aqueous and ethanol extracts of the plant “Vernonia mespilifolia” as an anti-obesity medicine. The authors claim that their results suggest that “Vernonia mespilifolia” is non-toxic and its acclaimed anti-obesity effects could be ascribed to its ability to inhibit both carbohydrate and fat digesting enzyme in the body.

The manuscript in the present form can not be accepted for publication because of the following reasons

The manuscript is not written well to describe each and every experiments that the authors performed with the observation followed by its meaning/interpretation. For example, there is no description in the main text about how the experiments is being done. Moreover, in the Figure 1, the authors used a control named “acarbose”, but there is no mention of this chemical and its use in the manuscript. As far as I understand, acarbose is a class of drugs that are alpha-glucosidase inhibitors, and I don’t know why the authors used acarbose as a control in the  α-amylase inhibition experiments. If this is not a mistake, authors should mention the details in the manuscript for the better understanding of the reader. Similarly, the same should be done for EGCG, the authors should describe why they used it as control and these should be cited properly using appropriate references. In the cytotoxicity experiments, I did not find any difference between the fluorescence images that are pasted in the manuscript. Either the authors should change the picture with a more clear one, or should red o the experiments with appropriate concentrations of the dyes to get a good image. I do accept and agree with the authors that the water extracts are non-toxic. But, you should note that the ethanol extracts are toxic as 80% of cells were killed at 200 microgram/liter of the ethanol extract. Hence, The statement in the discussion “From our results, it can be inferred that the aqueous and ethanolic extracts of V. mespilifolia did not display any cytotoxic nature in relation to the aforementioned statement. Therefore, it can be suggested that the aqueous and ethanolic extracts could be employed orally for the management of obesity” is not valid. The authors claim in the abstract that “acclaimed anti-obesity effects could be ascribed to its ability to inhibit both carbohydrate and fat digesting enzyme in the body”. But in the discussion, the authors said that “Our findings highlighted no significant inhibition of both amylase and glucosidase by both extracts in comparison to positive controls”. What is the meaning, please clarify. Are you stating that these materials are much less efficient than the controls? In the figures, the notations, a &b is not clear, please describe them clearly. Moreover, in the Figure1, there is a 33 and 3 are there in the last bar graphs, but these numbers are not explained anywhere. I recommend the authors to explain each and every data presented in the manuscript. Each bar graphs have a control point presented, what is the control used. Add this in either the text or in the figure caption. It is wrote in the discussion that “It should also be noted that both extracts moderate inhibition against amylase, however against α-glucosidase, the aqueous extract displayed superior inhibitory capability when compared with the ethanol extract (Table 1)”. I can see the difference in the Table 1, but not in the figure 2, where I can see that at lower concentration, ethanol extract showed much higher inhibitory activity, and the inhibitory activity equaled at higher concentrations for both water and ethanol extracts. Authors, please explain this difference and please describe how you ended up in such a difference in graph and IC50 values. Importantly, before making a strong statement like “Therefore, it can be suggested that the aqueous and ethanolic extracts could be employed orally for the management of obesity”, how the authors can assure or prove that these extracts/medicine does not have any side effects?

Overall, the manuscript in the present form should be modified thoroughly in terms of more control experiments and should be rewritten to describe each and every data in detail for the broader readership.

Reviewer 2 Report

The manuscript entitled “Inhibition of key enzymes linked to obesity and cytotoxic activities of whole plant extracts of Vernonia mesplilfoliaLess-A promising medicinal crop in South Africa” examined the effects of aqueous and ethanol extracts of Vernonia mesplilfoliafor inhibition of enzymes related to sugar uptake. Moreover, the cytotoxicity of the extracts against HeLa cells were determined. The authors described that the extracts are not toxic.

It is difficult to say in the present manuscript that the extracts are promising medicinal crop for preventing onset of obesity.

Comments:

1). In abstract, the authors described that the extracts were found non-toxic. However, in Figure 5 and Table 2, ethanol extract of Vernonia mesplilfoliashowed cytotoxicity to HeLa cells. The IC50value of the ethanol extract for cytotoxicity were 150 µg/mL that is lower than other IC50values for enzymatic activities in Table1. These results suggest that the ethanol extract of Vernonia mesplilfoliais highly toxic to human cells. Therefore, the ethanol extract is not promising to use for anti-obesity. How do the authors explain?

2). Why did the authors hypothesize that the extract of Vernonia mesplilfolia would be shown the inhibitory effects to the enzymes? Is that reported that the extract of Vernonia mesplilfolia showed anti-obesity activity in humans?

3). What are the meanings of a, and b in Figure 1, 2, and 3?

4). In Figure 3, it is difficult to see the purple bar (Ethanolic extract), because the bar is overlapping to words.

5). Please check the position of asterisks in Figure 5. At least in case for me, the asterisks are misaligned.

6). Please check the mistake of English in whole manuscript again. For example, “in vitro” in abstract (line 17) should be described in Italic.

Reviewer 3 Report

1. Which compounds are due to the inhibitory activity of these extracts? Nothing was presented regarding the chemical composition of the extracts, no correlations were made with certain compounds.

2. I understand that the extracts showed the highest percentage of inhibition on alpha glucosidase> alpha amylase> lipase (Fig. 1-3). In this context, the calculated value for lipase inhibition - IC50 = 331.16 mg / mL by the alcoholic extract (tab. 1) does not match the value of the lipase inhibition percentage in fig. 3 (the small IC50 value would mean a high inhibition percentage ).

3. Fig. 4 is not visible